# Review of Functional and Pharmacological Activities of Berries

**DOI:** 10.3390/molecules26133904

**Published:** 2021-06-25

**Authors:** Oksana Golovinskaia, Chin-Kun Wang

**Affiliations:** School of Nutrition, Chung Shan Medical University, 110, Section 1, Jianguo North Road, Taichung 40201, Taiwan; oksana2187@mail.ru

**Keywords:** berries, phytochemicals, bioavailability, pharmaceuticals properties, chronic diseases

## Abstract

Functional plant-based foods (such as fruits, vegetables, and berries) can improve health, have a preventive effect, and diminish the risk of different chronic diseases during in vivo and in vitro studies. Berries contain many phytochemicals, fibers, vitamins, and minerals. The primary phytochemicals in berry fruits are phenolic compounds including flavonoids (anthocyanins, flavonols, flavones, flavanols, flavanones, and isoflavonoids), tannins, and phenolic acids. Since berries have a high concentration of polyphenols, it is possible to use them for treating various diseases pharmacologically by acting on oxidative stress and inflammation, which are often the leading causes of diabetes, neurological, cardiovascular diseases, and cancer. This review examines commonly consumed berries: blackberries, blackcurrants, blueberries, cranberries, raspberries, black raspberries, and strawberries and their polyphenols as potential medicinal foods (due to the presence of pharmacologically active compounds) in the treatment of diabetes, cardiovascular problems, and other diseases. Moreover, much attention is paid to the bioavailability of active berry components. Hence, this comprehensive review shows that berries and their bioactive compounds possess medicinal properties and have therapeutic potential. Nevertheless, future clinical trials are required to study and improve the bioavailability of berries’ phenolic compounds and extend the evidence that the active compounds of berries can be used as medicinal foods against various diseases.

## 1. Introduction

Many studies and reviews have reported on the relationship between fruit intake and health. Some berries are currently used as ingredients in functional foods and dietary supplements. Berries are rich in nutrients and phytochemicals, which have been proven to improve health and prevent various chronic diseases during in vivo and in vitro studies. The primary phytochemicals in berry fruits are phenolic compounds including flavonoids (anthocyanins, flavonols, flavones, flavanols, flavanones, and isoflavonoids), tannins, and phenolic acids [1].

Raimundo and his co-workers [2] conducted a meta-analysis of a considerable number of different human randomized clinical trials in order to estimate the effects of polyphenol intake on biomarkers (such as the level of glucose, insulin, and others) in people with prediabetes and T2D. They found that the consumption of polyphenols may contribute to lower glucose levels.

Another meta-analysis of 128 randomized clinical trials was carried out in order to investigate the effects of plant sources of anthocyanins and ellagitannins (berries, nuts, red grapes/wine) on cardiometabolic risk biomarkers. Both anthocyanin and ellagitannin-containing products reduced total cholesterol. However, blood pressure was significantly decreased by the sources of anthocyanins such as berries and red grapes/wine. In contrast, the ellagitannin-containing products, especially nuts, were considerably effective in reducing waist circumference, LDL-cholesterol, triglycerides, and glucose [3].

Since berries have a high concentration of polyphenols, it is possible to use them for treating various diseases pharmacologically by acting on oxidative stress and inflammation, which are often the leading causes of diabetes, neurological, cardiovascular diseases, and cancer.

Blueberries have antioxidant and anti-inflammatory effects and also possess neurocognitive benefits. The consumption of blueberry juice improved memory function in older adults with early memory decline [4]. Black raspberries are sources of phenolic compounds such as ellagic acid and anthocyanins that have potential cancer chemopreventive activity, confirmed from the results of human clinical trials [5,6]. Blackcurrant powder reduced the activity of some colon cancer markers by acting as a prebiotic agent [7]. The antioxidant and anti-inflammatory properties of strawberries (due to their high content of bioactive compounds such as vitamins and phenols) has been displayed in several in vitro and in vivo studies [8,9].

This review examines commonly consumed berries: blackberries (*Rubus* sp.), blackcurrants (*Ribes nigrum*), blueberries (*Vaccinium* sp.), cranberries (*Vaccinium macrocarpon*), raspberries (*Rubus idaeus*), black raspberries (*Rubus occidentalis*), and strawberries (*Fragaria ananassa*) and their polyphenols as potential medicinal foods (due to the presence of pharmacologically active compounds) in the treatment of various diseases and disorders. The biologically active components of berries possess antioxidant, antihyperlipidemic, antihypertensive, and anti-proliferative effects and anti-inflammatory, antibacterial, and antiviral responses [10].

Polyphenols have a low bioavailability. Increasing their bioavailability can reduce the number of biotransformations of active compounds in the gastrointestinal tract and improve the health benefits of berries. This review discusses the studies conducted in vivo which consider the berries’ polyphenol bioavailability. Hence, this comprehensive review shows that berries and their bioactive compounds possess medicinal properties and therapeutic potential.

## 2. Data Collection

The authors of this comprehensive review article carried out a literature search for relevant articles regarding the functional and pharmacological activities of berries by determining sourced or literature in the form of primary data or official books and national or international journals published until May 2021. Additionally, data searches were also conducted using different online platforms. During the writing of this review article, the main references were cited from the trusted source such as Medline (PubMed), Scopus, Google Scholar, NCBI, Science Direct, ResearchGate, Web of Science and other trusted journals publishes with the following keywords: berries, phytochemicals, bioavailability, pharmaceuticals properties, and health benefits. This review article does not have any inclusion criteria, and PhD theses were included in the review. Furthermore, the search was only limited to articles published in the English language.

## 3. Composition

### 3.1. Nutrient Composition

Berries contain a large number of essential vitamins, dietary fibers, and minerals (Table 1). Berries are rich in sugars but are low in calories and lipids. Raspberries, blackberries, and blackcurrants contain vitamin C, dietary fibers, potassium, and folates. The levels of vitamin C range from 9.7 to 60 mg/100 g among these berries; blueberries have the lowest while strawberries have the highest. Strawberries, blackberries, and raspberries are excellent sources of folate (vitamin B9) and potassium. Cranberries are rich in vitamin E, and blackberries and blueberries contain high levels of vitamin K. Blackberries are a rich resource of beta-carotene, lutein, and zeaxanthin. Blackcurrants possess the most elevated levels of calcium, iron, phosphorus and potassium among these berries.

### 3.2. Phenolic Composition

The chemistry of berry phenolics affects their bioavailability, metabolism, and their biological effects in vivo. The total amount of phenolics, anthocyanins, and ellagic acid in berries are presented in Table 2 and Table 3. Berries also comprise condensed (nonhydrolyzable) tannins (known as proanthocyanidins), esters of gallic acid and ellagic acid (defined as hydrolyzable tannins), and stilbenes [12].

#### 3.2.1. Anthocyanins

Anthocyanins are natural pigments, accountable for the colors of many fruits and vegetables, which show antioxidant, anti-inflammatory and antimicrobial activities and also play an essential role in preventing diabetes, cancer, neuronal and cardiovascular diseases, etc. [37]. The anthocyanin levels in blueberries, blackberries, and black raspberries are much higher than in red raspberries, strawberries, and cranberries but are similar to blackcurrants. The most common anthocyanidins found in these berries are cyanidin, pelargonidin, delphinidin, malvidin, peonidin, and petunidin [38] Anthocyanins are formed by binding sugars to anthocyanidins (Figure 1).

#### 3.2.2. Proanthocynidins (PACs)

PACs are divided into several classes based on the hydroxylation of their constitutive units and the linkages between them (Figure 2). The most common constitutive units are (epi)catechin, (epi)gallocatechin, and, more infrequently, (epi)afzelechin. B-type PAC contains a single interflavan carbon bond linked through C4-C8 or C4-C6, while A-type PAC contains an additional interflavan bond linked through C2-O-C7 bonds [39].

Berries such as cranberries, blackcurrants, and blueberries are the best sources of proanthocyanidins (condensed tannins). The proanthocyanidins content varies with the level of ripening from red to black, reaching a maximum level but declining distinctly during the last stage of ripening. Research on blackberries shows that the contents of proanthocyanidins and anthocyanins vary from different growth stages [40]. Proanthocyanidins give astringency, sourness, bitterness, sweetness, saliva viscosity, aroma, and color composition. Both cranberries and blueberries are exceptionally rich in PACs (Table 4). Cranberries have a higher PACs content when compared with other berries. PACs blueberries are slightly lower. PACs show antioxidant, anti-inflammatory, antibacterial, antiviral, anti-carcinogenic, and vasodilatory effects [41].

#### 3.2.3. Flavonols

Flavonols are phenolic compounds (kaempferol, quercetin, and myricetin) (Figure 3 and Table 5), poorly soluble substances present in berries which possess antioxidant activity, anticancer and antibacterial properties, and protect against cardiovascular disorders [45].

Blueberries are good sources of flavonols, especially quercetin and myricetin [53]. Quercetin is the major flavonol in cranberries and black raspberries. Myricetin glycosides are also presents in these berries but in lesser quantity [50]. Blackberries contain nine quercetin and three kaempferol derivatives [54]. Quercetin and kaempferol are present in red raspberries [55].

#### 3.2.4. Phenolic Acid

Hydroxycinnamic acids such as p-coumaric, caffeic, ferulic acids, and hydroxybenzoic acids such as p-hydroxybenzoic, gallic, and ellagic acids show antioxidation and anticancer effects [56]. Cranberries contain notable quantities of ursolic acid in its peel (in the aglycone form), and also varieties of phenolic acids (among which the major one is p-hydroxycinnamic acid) [57]. Blueberries contain gallic acid, caffeic acid, and ferulic acids. One of the essential phenolics of blueberries is chlorogenic acid, which has powerful antioxidant properties [58]. Red raspberries contain hydroxycinnamic acids (caffeic, p-coumaric, and ferulic acids), hydroxybenzoic acids (ellagic and p-hydroxybenzoic acids) [55]. Strawberries are extremely rich in ellagic acid. Ellagic acid in strawberry exists at a free form and is esterified to glucose at water-soluble hydrolyzable ellagitannins [59]. The total ellagic acid contents, determined after acid hydrolysis, are 319.7 mg/100 g in black raspberries. Ellagic acid displays a wide range of biological properties such as radical scavenging, cancer prevention, and anti-inflammatory and antibacterial effects [31].

#### 3.2.5. Ellagitannins

Various studies indicate dietary ellagitannins or ellagic acid may have beneficial impacts on health. Ellagitannins are complex derivatives of ellagic acid and belong to the class of hydrolyzable tannins, showing antioxidant, antimicrobial, anti-inflammatory, anticarcinogenic, and anti-*Helicobacterpylori* (*H. pylori*) properties. Ellagitannins are abundant in strawberries, raspberries, and blackberries [60]. Ellagitannins are the prominent hydrolyzable tannins in blackberry, along with sanguiin H-6 and lambertianin. The main ellagitannins in raspberries are the sanguiin H-6 dimer and the C lambertian trimer [61]. The metabolism of ellagitannins is shown in Figure 4.

#### 3.2.6. Stilbenes

Stilbenes are a specific class of non-flavonoid phenolic compounds present in berries. The most popular compound identified in berries is resveratrol. Grapes and red wine are among the primary dietary sources of stilbenes. Stilbenes possess different biological and pharmacological activities which potentially beneficial for human health such as neuroprotective, antitumor, and antioxidant effects. Berries such as blueberries and cranberries contain stilbenes [62,63].

## 4. Bioavailability

### 4.1. Anthocyanins

Many anthocyanins appear in urine after consuming berries, albeit in low concentrations, around 0.1% or less, of the ingested dose [64]. Anthocyanins are found in human plasma in low concentrations 0.5–1 h after eating. Thus, unlike flavonol glycosides, glycosylated anthocyanins appear in the bloodstream. It may be a result of the fact that anthocyanin glucosides are not hydrolyzed by human small intestinal β-glucosidases, unlike quercetin glucosides [65]. Studies conducted on rats show that anthocyanin absorption occurs in the stomach and the small intestine. Their absorption from the stomach into the blood may explain their rapid but temporary increase in serum [66]. Anthocyanins are associate with bilitranslocase, providing a plausible mechanism for absorption from the stomach [67]. Volunteers took freeze-dried black raspberries every day for seven days in the amount of 45 g of powder, which is equivalent to two cups of fresh product and contains 15–20 mg/g of anthocyanins. After ingestion, anthocyanin levels peaked in plasma between 1 and 2 h and in urine during 0–4 and 4–8 h of collection [68]. The intake of 300 g of raspberries, which are low in polyphenols, displayed anthocyanin metabolites excreted in the urine in amounts of 15% of total consumption [69]. Human studies illustrate that the time to reach the maximum plasma concentration of anthocyanins is from 0.5 to 4 h, which is compatible with researches showing that anthocyanins can be partially absorbed in the stomach before entering the small intestine [64,70]. The bioavailability of the significant anthocyanin of strawberries and blackberries is pelargonidin-3-glucoside or cyanidin-3-glucoside, respectively (reviewed in humans). The preponderance of compounds recovered are monoglucuronide metabolites. It is also is found methylated glycosides and sulfoconjugates, albeit in small quantities. Aglycone structure may play a significant role in bioavailability, as pelargonidin has a total mean recovery of 1.80% and cyanidin has only 0.16% of the consumed amount [71,72]. After consuming blackcurrant, about 73% of its anthocyanins entered the colon and were metabolized by microorganisms [73]. The primary metabolite of anthocyanins found in urine of rats after eating blueberries is hippuric acid [74]. Phenolic acids of anthocyanins are absorbed in the colon. They are possibly further metabolized by the liver [75]. Unabsorbed anthocyanins enter the colon and may be converted to other metabolites by colonic bacteria, followed by absorption or excretion in feces. Colonic microbiota hydrolyses glycosides into aglycones and degrades them into simple phenolic acids [76]. Low levels of glycosylated anthocyanins are directly absorbed in the small intestine. The intestinal microbiota hydrolyzes anthocyanins by β-glucosidase. The resulting aglycones are metabolized into various phenolic and aldehyde components [77].

### 4.2. Proanthocyanidins

The bioavailability of proanthocyanidins is primarily influenced by the degree of polymerization. Proanthocyanidins in the gastrointestinal tract have insignificant depolymerization. The majority of proanthocyanidins reach the colon intact and are degraded into phenylvalerolactones and phenolic acids by colon microbiota. These microbial metabolites may provide the health-promoting attributes of proanthocyanidins in vivo [78].

### 4.3. Flavonols

Anthocyanins appear in plasma and are excreted in the urine in considerably more diminutive concentrations than flavonols. Quercetin from berries is bioavailable. Quercetin levels in plasma increase to 50% in subjects when they consumed 100 g/day of bilberries, lingonberries, and blackcurrants for two months [79]. Nevertheless, another study does not recover quercetin or myricetin in volunteer’s plasma following an acute product dose of compounds from cranberry juice and fruits [80]. Most of the other phenolics are retrieved in the plasma and have smaller molecular weights. Some of the phenolics had two absorption peaks, indicating that reabsorption of the compounds eliminated in the bile or the metabolism of high molecular weight are not absorbed in the stomach by the gut microbiota [81].

### 4.4. Phenolic Acids

Phenolic acids recovered after eating black raspberries are extracted from the gastrointestinal tract. Protocatechuic acid and 3-hydroxybenzoic acid are obtained significantly than their initial level, but p-coumaric acid, ferulic acid, and caffeic acid are in less quantities. This indicates the production of phenolic acids from various sources in the gastrointestinal tract. Phenolic acids are determined in rat’s urine after eating cranberries, blueberries, or blackberries, and phenolic acids are found in both free and conjugated forms [82]. Cranberries are rich in proanthocyanidins, mainly resulting in the formation of 4-hydroxycinnamic acid. After eating blueberries, which also contain many proanthocyanidins, chlorogenic, ferulic, and 3,4-hydroxycinnamic acids were found in rat’s urine. Black raspberries, which have primary cyanidines, 3-hydroxyphenylpropionic, 3-hydroxybenzoic, and 3-hydroxycinnamic acids, were also found after intake [83].

### 4.5. Ellagitannins

Ellagitannins are present in high levels in raspberries, blackberries, and strawberries and have high molecular weight (which is too large to be absorbed). They can depolymerize into gallic acid and ellagic acid, which are better absorbed [84]. Ellagitannins from raspberry juice (lambertianine C and sanguine H-6) are not extracted in the gastrointestinal tract of rats either in the entire gastrointestinal tract or in plasma, urine, or feces 1 h after ingestion. The acidic pH conditions are possibly responsible for the fast breakdown of these molecules. Ellagic acid is recovered in the stomach (9.6% of its original amount in juice) but not in blood and plasma [85]. Ellagitannin metabolites are found in the plasma of mice, liver, prostate, and colon in the form of urolithin A and C, produced by mice’s microbiota after consuming black raspberries. Ellagitannins are partially hydrolyzed in the mice intestine and release ellagic acid [86]. Another study shows that colon microbiota could metabolize ellagic acid to form urolithin B. Ellagic acid and urolithin B are absorbed by humans and determined in blood and urine samples [87]. The excretion of ellagic acid and its derivatives in humans has been studied after consuming strawberries, raspberries, walnuts, or red wine. Urine samples were collected at 8, 16, 32, 40, and 56 h after oral administration. Ellagic acid was not detected in urine samples. Nevertheless, urolithin B conjugated with glucuronic acid, was rich in all groups except the control one [87]. Ellagitannins release extrication ellagic acid in vivo, followed by the metabolism of and production of urolithins D, C, A, and B (in that order) from the jejunum to the distal portion of the pig’s intestine. Moreover, the absorption of these metabolites was observed to increase with their increasing lipophilicity. Glucuronides, methyl glucuronides, and urolithins of ellagic acid were detected in the bile and plasma of pigs [54,88]. The urolithin metabolites were excreted in the urine for much more extended periods than the anthocyanin metabolites [69]. Urolithins A and B were found in peripheral plasma. The presence of ellagic acid metabolites in bile and urine and its absence in intestinal tissues suggest its absorption in the stomach, and it will explain the results on the absorption of free ellagic acid within short periods (from 30 min to 1 h) after ingestion. Urolithin A is the only metabolite found in feces and, together with glucuronide, is the most abundant metabolite in urine. The metabolites are not accumulated in any analyzed organ [89].

It is necessary to improve the bioavailability of phenolic compounds by using various methods of increasing the stability and solubility in the gastrointestinal tract such as selective inclusion, solid dispersion, phospholipid liposomes, microemulsion technology, and the conversion of flavonoid aglycones into nanoparticles [90,91,92,93,94].

## 5. Oxidative Stress Suppression

The formation of high amounts of free radicals may generate oxidative stress, leading to many degenerative disorders and aging. The antioxidant property of berries connects with active oxygen radical scavengers such as vitamin C, phenolic compounds, and carotenoids. The antioxidant capacity of berries is four times higher than other fruits and ten times higher than vegetables. Studies have confirmed that strawberries’ antioxidant ability closely correlates with their potent phenolic compounds and vitamin C content [58]. Vitamin C is the essential antioxidant component in strawberries, followed by anthocyanins and then hydroxycinnamic acids (mainly p-coumaric acid derivatives and flavanols) [95]. Strawberry consumption increases plasma antioxidant capacity, reduces oxidative damage of plasma proteins and increases vitamin C levels in the serum [96,97]. Moreover, strawberry extracts prevent ethanol-induced gastric damage in vivo. Therefore, a diet rich in strawberries positively improves gastric diseases caused by oxidative damage [98]. Raspberries exhibit many antioxidant components and possess high radical scavenging activity. Lyophilized aqueous extracts of different types of raspberries contain p-coumaric acid, which is mainly responsible for free radical scavenging of raspberries [99]. Anthocyanins, ellagitannins, and polyphenols also showed antioxidant and tumor proliferation inhibitory activities [100]. Studies show that the 75% antioxidant capacity of raspberries is associated with anthocyanins and ellagitannins [101]. Cyanidin 3-rutinoside and cyanidin 3-xylosylrutinoside (Figure 5) are found in black raspberries in the highest concentration and are its primary antioxidants which, together with the other bioactive constituents of black raspberries, show potential biological activity in clinical trials for the therapy of various types of cancer [102].

Blackberries are rich in antioxidants that reduce oxidative stress and have a high oxygen radical absorbance capacity (ORAC) [54]. Blackberries and black raspberries show higher antioxidant activities than red raspberries and contain high amounts of cyanidin glycosides, which are potent antioxidants. More than that, blackberries and strawberries have the highest ORAC during the green stage than the ripe stage [103]. The ORAC value for the berries is presented in Table 6.

Blueberries contain high amounts of polyphenols, procyanidins, and anthocyanins, have high antioxidant properties, and reduce oxidative stress. They work as radical scavengers and help prevent various diseases, including cancer [105]. The antioxidant capability of blueberries is higher in the early ripening set than in ripe berries due to the higher concentrations of hydroxycinnamic acid and flavonols in immature berries [106]. Hydroxycinnamic acids and anthocyanins of blueberries effectively reduce oxidative stress in endothelial cells and ovarian, murine melanoma, and cervical cancer cell lines [107,108,109]. In addition, inscribed significant increases in the hydrophilic and lipophilic antioxidant capacity of human plasma following the ingestion of blueberries [110]. The antioxidant properties of blackcurrant are mainly associated with anthocyanins [111]. Nour and his co-workers [112] indicated a high correlation between antioxidant activity and the total concentration of anthocyanins. Phenolics are a significant contributor to antioxidant activity in blackcurrants, but vitamin C also makes an essential contribution to antioxidant activity. Blackcurrant is high in ascorbic acid (50 to 280 mg/100 g) and flavonoids content, which increases both the antioxidant capacity of the berries and their potential to help health benefits [113]. Human and animal studies have shown the effects of blackcurrants on athletic training and performance. Blackcurrants lower oxidative stress-related injuries that can cause fatigue and damage. Thus, blackcurrants are likely an essential source of antioxidants for the human diet [114]. Cranberries are one of the best antioxidants among berries due to phytochemicals such as benzoic and cinnamic acid derivatives and flavonols. The anthocyanins and flavanols of cranberries can inhibit oxidative damage induced by ROS and have higher radical scavenging activity than vitamin E. Cranberry bioactive compounds have antioxidant effects both in vitro and in vivo. The PACs of cranberries help inhibit oxidative stress and possess antibacterial properties [115]. Consumption of cranberry juice has improved plasma antioxidant capacity, and thereby decreased the circulating concentrations oxidized low-density lipoprotein (LDL) cholesterol in women with metabolic syndrome, as well as decreased blood markers of oxidative stress in healthy volunteers and patients with cardiovascular risk factors [116,117].

## 6. Antimicrobial Properties

Many plants produce antimicrobial secondary metabolites as part of their normal growth process and respond to pathogen attacks. For example, *H. pylori*-infected subjects consumed with the burdock complex for eight weeks had significantly lowered urea breath test values and inflammatory markers compared to the placebo. Studies support the high antimicrobial potential of plant extracts [118,119]. Blackcurrant juice has antimicrobial properties, and its anthocyanins inhibit the adhesion of *Typhimurium Salmonella* to human epithelial colorectal adenocarcinoma cells (Caco-2) by up to 39% [73]. Polysaccharides from blackcurrant seed extracts are found to inhibit *H. pylori* adhesion to the human gastric mucosa [120]. Phenolic extracts (concentration 1 mg/mL) from blueberries, blackcurrants, raspberries, and strawberries can inhibit the growth of *H. pylori* [121]. Ellagitannins of red raspberries inhibit the growth of human pathogenic bacteria strains such as *Salmonella* and *Staphylococcus* [122]. Extracts of cranberry inhibit the adhesion of uropathogenic *Escherichia coli* (*E. coli*), as well as the interference of *H. pylori* to the human gastric mucosa [123,124]. Daily consumption of cranberry juice has been found to fight against *H. pylori* infections and has shown significant bacterial suppression in clinical trials [125]. Animal studies show that berry components such as epicatechin, chlorogenic acid and quercetin effectively counteract nonsteroidal anti-inflammatory drugs mediated damage and *H. pylori* infection. The effect of berries on *H. pylori* infection is also approved in human trials by using cranberries, which contains active fractions of quercetin and epicatechin [50]. The randomized, controlled, double-blind multicentric trial carried out in 295 asymptomatic children positive for *H. pylori* shows that regular or frequent cranberry juice consumption could be helpful therapy in asymptomatic children with *H. pylori* infections [126]. Cranberry fruit and its extracts inhibit *H. pylori* and show anti-adhesive, anti-inflammatory, and antibacterial bioactivities [127]. A randomized placebo-controlled clinical study shows that intake of cranberry extract containing A-type PACs reduces bacterial adhesion *E. coli* and prevents urinary tract infection [128]. A-type PACs trimers of cranberry are more effective inhibitors than A-type PACs dimers, while B-type PACs do not work [129]. B-type PACs from blueberries decrease avian influenza virus (AiV) titers with the potential to prevent AiV infections and relieve the illness symptoms connected with AiV infections [130]. A nondialyzable fraction of cranberry juice contains about 65% of PACs and decreases the bacteria’s adhesion properties. Therefore, this fraction has been added to mouthwash and its effects on oral hygiene have been investigated. After six weeks of treatment, there were significantly decreased numbers of salivary *Streptococci mutans* and total bacteria [131]. Blueberries affect the number of microbes in the cecum and reduce the number of *Clostridium perfringens*, *Enterococcus*, and *E. coli*, which are associated with irritable bowel syndrome [132]. Rats, fed 4 mL/kg of blueberry extract daily for six consecutive days, showed a substantial rise in the number of *Lactobacilli* and *Bifidobacteria*, demonstrating the prebiotic potential of blueberries [133].

## 7. Anticancer Properties

Studies in vitro and in vivo show that the antioxidant effect of berries correlates with its anticancer potential [134,135]. The antioxidant mechanism includes scavenging reactive oxygen species (ROS) that induce oxidative damage to cellular macromolecules such as DNA and RNA. The accumulation of oxidative DNA damage contributes to the formation of tumors and, thus, oxidative stress represents one of the significant reasons for enhancing carcinogenesis [136]. The extract of strawberries significantly decrease tumor volume and increase the mice model’s lifespan [137]. Bioactive compounds in strawberries inhibited azoxymethane/dextran sodium sulfate-induced colon carcinogenesis in a murine model [138]. Moreover, an in vitro study showed anti-proliferative effect of strawberry extracts with ellagic acid on the human colon, prostate, and oral cell lines [139]. An intake of 60 g of freeze-dried strawberry powder per day decreased the histological grade of premalignant lesions and levels of various pro-inflammatory proteins in more than 80% of patients with esophageal dysplasia [140]. Strawberry and black raspberry extracts show perfect pro-apoptotic effect on the HT-29 colon cancer cell line, which expresses cyclooxygenase 2 (COX-2). The extracts induced apoptosis three times when compared to untreated control ones. Human oral, breast, lung, prostate, and colon cancer cell lines treated with the other berry extracts (blueberry, blackberry, and red raspberry) also show increased apoptosis levels in 1.8 times at compared with control [141,142]. Both anthocyanins and ellagic acid show anti-proliferative activities across many different human cancer cell lines. Black raspberries are rich in ellagitannins and anthocyanins, which show chemopreventive potential. Studies show that freeze-dried black raspberries reduce carcinogen-induced colon and esophageal carcinogenesis in animals [143,144]. The effect of berries on esophageal diseases was carried out in mice model, which were injected with N -nitrosomethylbenzylamine (NMBA) three times a week for five weeks, after which they were fed a diet with 5% berry content. Black raspberry feeding reduced dysplastic lesions and significantly decreased the mean papilloma size [145]. Similar decreases in the size and number of tumors are shown by blueberries, strawberries, red raspberries, and blackberries [146]. After six weeks of ingestion (four times a day), black raspberries, in the form of a bioadhesive gel, significantly decreased COX-2 protein levels and inhibited the apoptosis in patients with precancerous lesions of the oral cavity [147]. This suggests that these anti-tumor actions governed by flavonoids, such as quercetin, are present in excess in black raspberries [148]. The oral consumption of 45 g per day of black raspberry powder by humans did not reduce the segment length of Barrett’s lesions in 26 weeks of study. Nevertheless, daily consumption of black raspberry promoted reductions in the urinary excretion of two oxidative stress markers in patients with Barrett’s esophagus [149]. Other clinical trials confirmed that black raspberry demethylates tumor suppressor genes and modulates other tumor development biomarkers in the human colon and rectum [150]. Patients with colorectal cancer had an intake 60 g black raspberries powder daily for nine weeks. In addition, urine and plasma specimens were collected before and after black raspberries intervention. Consumption of black raspberries led to significant changes in metabolites derived from black raspberries components both in urine and plasma. Furthermore, the correlation between these metabolites and tumor markers implies that black raspberries derived metabolites may provide beneficial regulation against colorectal tumors [5]. Thus, the treatment of colorectal cancer patients with berries may be beneficial and used in anti-cancer therapy. Li-Shu Wang and co-workers conducted clinical trials and discovered that black raspberries suppositories can suppress rectal polyp development in patients with familial adenomatous polyposis [6]. Mice with esophageal adenocarcinoma were injected with 250 µg of cranberry-derived proanthocyanidin extract and after three weeks showed significantly lower tumor volumes [151]. The PACs of cranberry reduced the number of stomach cancer cells both in vitro and in vivo [152]. Mice treated with cranberry extracts decreased intestinal inflammation and blood circulation lipopolysaccharides (LPS) [153]. Blueberry extracts are able to completely inhibit the proliferation of several tumor cell lines in vitro including the colon (HT-29 and HCT116), prostate (LNCaP), breast, mouth (KB and CAL-27), cervix (HeLa), ovaries (A2780), and skin (B16F10) [107]. Consuming a diet rich in blueberries may be effective against estrogen-mediated breast cancer [154]. Rats fed a diet containing residual fractions of berries (blueberries, raspberries, and strawberries) reduced NMBA-induced esophageal carcinogenesis irrespective of their ellagitannin content [155]. Blackberry anthocyanins impeded cancer cells growth by modifying cellular signaling pathways such as modulating the expression of activating protein-1 (AP-1) and nuclear factor-kB (NF-kB), essential proteins that coordinate cell proliferation, vascular endothelial growth factor, and COX-2 [156]. Furthermore, quercetin extracted from blackberries showed anti-carcinogenic properties in animal models and human carcinoma cell lines (HT29 and Caco-2) [157].

## 8. Diabetes

Berries rich in biologically active phytochemicals, particularly anthocyanins and proanthocyanidins, can suppress the rise in blood glucose levels, improve diabetes and other metabolic disorders [158]. The supposed mechanism for decreasing postprandial glucose is to limit glucose absorption by inhibiting α-amylase and α-glucosidase activity. Compare with other extracts of berries, red raspberry extracts are the most effective in inhibiting α-amylase. Raspberry extract fractionation reveals that the unbound anthocyanin-enriched fraction is more effective against α-glucosidase than the original extract. In contrast, the α-amylase inhibitors were accumulating in the bound fraction. This suggest that proanthocyanidins are important inhibitors of α-amylase activity [159,160]. The other study shows that anthocyanins improve the function of adipocytes and enhance the insulin sensitivity [161]. Supplementation of red raspberry ellagic acid at 2% and 5% of the diet for 12 weeks increased insulin levels and reduced fasting glucose, hemoglobin, and glycosylated urinary albumin, thereby improving the uncontrolled diabetic status of mice. Indicators of inflammation and oxidative stress were also improved [162]. Anthocyanins and ellagic acid show antioxidant, anti-inflammatory actions, and the potential for insulin secretion from pancreatic β-cells, which is found in cell culture [163], in diabetic animals [162,164], and in humans [165,166]. Strawberries inhibit against glucosidase and angiotensin-1-converting enzymes and have a less pronounced potential to inhibit a-amylase [167]. Blackcurrant extracts can decrease blood glucose and improve glucose tolerance in type 2 diabetic (T2D) mice and humans [168]. The anthocyanins of blueberry are shown to attenuate insulin sensitivity and hyperglycemia. Diet supplemented with blueberry powder enhances glucose tolerance in mice, normalizes glucose metabolism markers in obese rats, and improves insulin sensitivity in humans [169,170]. Furthermore, anthocyanins have been confirmed to induce glucagon production like peptide-1, which associates with pancreatic cells responsible for the induction of insulin secretion [171]. The consumption of cranberries, a natural source of polyphenols and fibers, would enable a more favorable glycemic response in patients with T2D. Cranberry contain the soluble fibers polydextrose and β-glucan, which have been related to the reduction in the rate of gastric glucose absorption. The flavonoids of cranberries delay the intestinal absorption of glucose and improve glycemic response [172,173]. The extent of inhibition of α-glucosidase by berry extracts is related to its anthocyanin content-cyanidin-3-rutinoside and cyanidin-3-galactoside. Proanthocyanidins are also considered potent α-glucosidase inhibitors [159]. Tannins (proanthocyanidins and ellagitannins) of cranberry extracts can inhibit digestive enzymes α-amylase and glucoamylase [174]. More than that, cranberry procyanidins can inhibit the glycation of human hemoglobin and serum albumin by eliminating reactive carbonyl radicals [175]. Some studies show that cranberry products might promote glucose homeostasis by reducing fasting glycemia, improving homeostasis model assessment-estimated insulin resistance, increasing insulin sensitivity, and preventing compensatory insulin secretion [176,177].

## 9. Obesity

Obesity is complicated to treat and is a significant risk factor for some health problems such as diabetes and cardiovascular disease. Since drug treatment of obesity induces many side effects and has little long-term efficacy, natural plant extracts have been suggested to use as an alternative for long-term weight control [178]. Cyanidin 3-glycoside is the predominant anthocyanin in blackberries found to prevent obesity in C57BL/6J mice fed a high-fat diet correlated to mice fed a high-fat diet without anthocyanins [179]. Blueberry juice and freeze-dried blueberries did not significantly affect weight gain or fat accumulation in mice fed a high-fat diet. However, blueberries anthocyanin extracts significantly reduced body weight and fat accumulation [180]. The anthocyanins of blueberry stimulated the transcription of the peroxisome proliferator-activated receptor (PPAR, participate in energy homeostasis regulation), which is associated with improving insulin resistance and fat stimulation metabolism in combination with inhibition of fat storage [181]. A human study observed improvement in the lipid profile and inflammatory markers in obese subjects after a three week intake of strawberry powder [182]. Another study shows consumption of freeze-dried strawberry for 12 weeks was able to improve the inflammatory condition in obese adults with osteoarthritis, lowered tumor nuclear factor-α, and lipid peroxidation products [183]. Strawberry consumption decreased risk factors for cardiovascular disease and diabetes in obese volunteers, offering a therapeutic potential for strawberries as a medicinal food to reduce obesity-related disease.

## 10. Cardiovascular Disease

Cardiovascular disease (CD) and cancer result from continued exposure to oxidative stress (OS). Reactive oxygen species (ROS) play an essential role in developing cardiovascular diseases and cancer since ROS induced by OS leads to apoptosis and necrosis. Anthocyanins show significant ROS trapping and lowering activity and can decrease DNA damage to protect the body [184,185] (for example, the main anthocyanins of blueberry-malvidin can decrease the concentration of ROS) [186]. Berries protect the body from cardiovascular disease through restraining platelet aggregation, affecting blood lipids, lowering OS, improving endothelial function, and regulating metabolism [187]. Anthocyanins significantly decrease the area of atherosclerotic plaque, mitigate the damage to endothelial cells and elastic plate, and diminish the number of foam cells and vascular wall proliferation [188]. Rodriguez-Mateos and co-workers provided evidence that in healthy humans, blueberry consumption leads to cardiovascular benefits (i.e., improvements in vascular function) linked with anthocyanin metabolites [189]. Cranberry vinegar shows a protective effect against CD, repressing in vivo oxidation of LDL (advanced plasma LDL levels are a risk factor for CD) by suppressing free radicals and donating hydrogen atoms [190]. A double-blind, randomized controlled crossover trial displayed dose-dependent vascular function improvements in healthy males after consuming cranberry juices [191]. The high contents of cyanidin glycosides in blackberries are responsible for increasing antioxidant activity and protection against LDL oxidation, while anthocyanins, flavan-3-ols, and hydroxycinnamic acids inhibit liposomal oxidation [13,192]. Most berries contain flavonoids which improve blood flow, endothelial function, and reduce the risk of CD [193]. With the development of vascular diseases, vascular endothelial cells (VECs) are damaged and can integrate anthocyanins into their membrane and cytosol. Anthocyanins help maintain VECs function by stabilizing the cell membrane or keeping oxidative balance [194]. 

## 11. Blood Pressure

Polyphenols and flavonoids are beneficial in the treatment of cardiovascular diseases, including hypertension. Some studies show a significant blood-pressure-lowering activity related to anthocyanins and anthocyanin-rich berry consumption [195,196]. Intake of a blackcurrant extract providing either 105, 210, or 315 mg/day of anthocyanins, after 12 days, display a significant decrease in arterial pressure in a group of 15 athletes with the two higher anthocyanins doses [197]. The black raspberry powder got to 45 prehypertensive subjects during eight weeks and a notable decline in systolic blood pressure (SBP) was noticed in the group receiving the 2.5 g raspberry power per day [198]. Zhu and his co-workers [199] carried a meta-analysis of different randomized clinical trials and discovered no notable blueberry consumption effect on either systolic or diastolic blood pressure. The other study involved elderly subjects who consumed whole wild blueberry powder (1 to 2 g per day) or 200 mg of blueberry extract (2.7, 5.4, or 14 mg anthocyanin content) for six weeks. A significant decrease in SBP was observed with the extract providing a higher dose of anthocyanins but not whole berry powder [200].

## 12. Neuroprotection

Age-related neurological diseases in many countries are increasing, and two of the most destructive are Alzheimer’s disease (AD) and Parkinson’s diseases (PD). The exact mechanisms of AD are unknown, but much scientific evidence suggests that oxidative stress, including free radicals, plays a crucial role in AD. The cytoplasm of vulnerable neurons is the leading site of increased oxidative damage. Also, inflammation may play an essential role in neuronal damage in AD’s early stages [201,202]. Many studies on the effects of polyphenol-rich plants (i.e., green tea, curcumin, apple, blueberry, strawberry, pomegranate, and cocoa) on neurological diseases suggests many potential mechanisms of polyphenol action in neuroprotection against oxidative stress [203]. Additionally, flavonols, especially quercetin and its derivatives and naringenin, from many plants (for example, strawberries, grapes, blackcurrants, green tea, citrus fruits, and cocoa) inhibit the formation of ROS caused by beta-amyloid protein and thus reduce oxidative stress-induced damage to nerve cell membranes more effectively than vitamin C [204]. Anthocyanins are the most potent neuroprotective phenolic compounds found in soft fruits. Blackcurrant contains various flavonols, including high amounts of myricetin, quercetin, and isorhamnetin, which show neuroprotective activity [205]. Phenolic extracts of blackcurrants also show effective neuroprotection against oxidative stress-induced neuronal damage in human cell cultures [206]. Several human studies indicate alterations to cognitive performance, modulation of blood flow, and control of blood glucose related to normal cognitive function after eating blackcurrant [207,208]. Blackberries have positive effects on age-related changes and may be profitable for preventing age-related neurodegenerative diseases such as AD and PD. Fischer’s rats, fed a diet supplemented with 2% blackberries, showed an improvement in motor activity when performing three tasks for balance, coordination, and cognitive abilities [209]. Consumption of blueberries positively affects brain function changes associated with age and caused by oxidative stress [4]. Blueberry intake has a neuroprotective effect in vitro against damage induced by a difference of neurotoxic agents and exhibits some in vivo protective and even advanced learning and memory abilities in mice [210]. Cranberry extract represses the oxidation of the neurotransmitter 6-hydroxydopamine in a cell model of Parkinson’s disease [211]. Aged rats given 2% (by weight) of freeze-dried cranberry show greater strength and balance than controls in motor skills tests. Brain tissue from the cranberry group show improved nerve signaling and a better response to oxidative stress ex vivo after 16 weeks of supplementation. Thus, cranberries improve neural function, neuroprotective responses, and some motor functions in aged animals [212]. Pelargonidin-specific anthocyanidin in strawberries inhibits proteasome activity and has neuroprotective effects [213]. Devore and his co-workers [214] studied the relationship between long-term consumption of strawberries and decreased cognitive function and found that a slower rate of decline in cognitive function correlates with more frequent consumption of strawberries for the elderly. The polyphenolic components of red raspberries reduced oxidative stress, inflammation, improve insulin signaling, and effectively reduced the risk of Alzheimer’s disease and slowed the aging process [215]. Feeding rats 100 mg/kg of Ellagic acid per day during seven days before traumatic brain injury significantly prevented memory impairment [216]. Anthocyanins protect against inflammatory and oxidative stress-mediated neuroinflammation and neurodegeneration in the cerebral cortex of adult mice [217], as well as in the brains of postnatal rats [218] and rats fed a high-fat diet [219]. Prepared anthocyanin-free (ACFs) and anthocyanin-enriched (ACEs) extracts from crude berry extracts (blackberries, black raspberries, blueberries, cranberries, red raspberries, and strawberries) were used to determine anthocyanins are significant contributor to neuroprotective effects. The berry ACEs presented excellent antioxidant, methylglyoxal (MGO) trapping, and anti-glycation activities when compared with their respective crude extracts (CEs) and ACFs. When MGO induced protein glycation, the berry ACEs show significant inhibition against the formation of glycation endproducts when compared with their respective CEs and ACFs [215]. Anthocyanins, such as delphinidin-3-rutinoside and cyanidin-3-O-rutinoside from blackcurrant berry extract trap MGO by forming anthocyanins-mono-MGO adducts [220]. Berries have potential neuroprotective effects, and their anthocyanins could contribute to these biological effects.

## 13. Conclusions

Berries are excellent sources of bioactive compounds that show notable health benefits as reported in both in vitro and in vivo studies. They contain potent antioxidants and exert protective effects against inflammatory disorders, metabolic disorders, cardiovascular diseases, and can suppress the risk of various cancers. They also possess antimicrobial and neuroprotective properties. The interaction between berry phenolics and the microbiota plays an essential role in berry phenolics bioavailability and contributes to gut health. Berries are potential pharmaceutical agents for the treatment of many diseases. Future clinical trials are required to study and improve the bioavailability of the phenolic compounds of berries and extend the evidence that the active compounds of berries can be used as medicinal foods against various diseases.

## Figures and Tables

**Figure 1 molecules-26-03904-f001:**
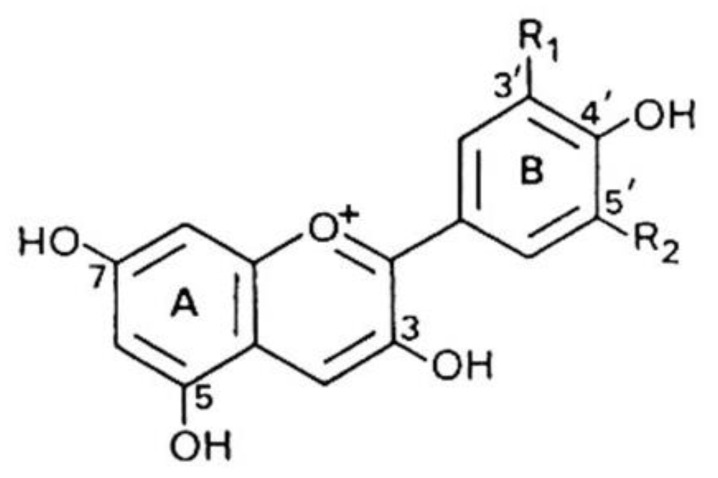
Chemical structure of anthocyanidins: R_1_ = H, R_2_ = H Pelargonidin; R_1_ = OH, R_2_ = H Cyanidin; R_1_ = OH, R_2_ = OH Delphinidin; R_1_ = OCH_3_, R_2_ = H Peonidin; R_1_ = OCH_3_, R_2_ = OH Petunidin; R_1_ = OCH_3_, R_2_ = OCH_3_ Malvidin.

**Figure 2 molecules-26-03904-f002:**
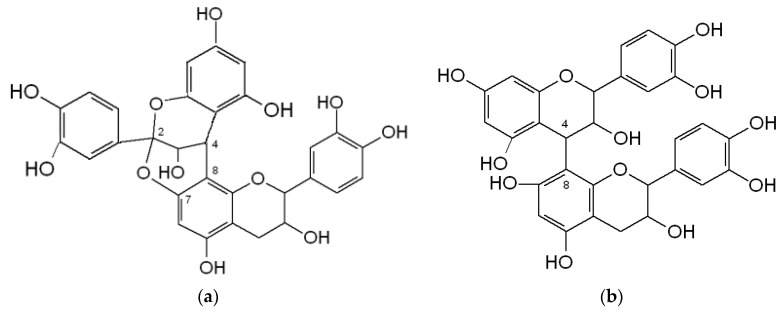
Chemical structure: (**a**) A-type proanthocyanidins; (**b**) B-type proanthocyanidins.

**Figure 3 molecules-26-03904-f003:**
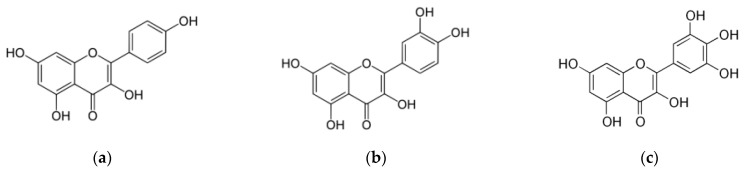
Chemical structures: (**a**) Kaempferol; (**b**) Quercetin; and (**c**) Myricetin.

**Figure 4 molecules-26-03904-f004:**
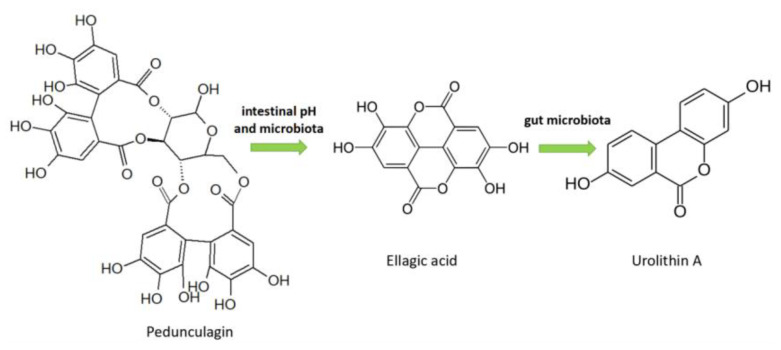
Metabolism of ellagitannins.

**Figure 5 molecules-26-03904-f005:**
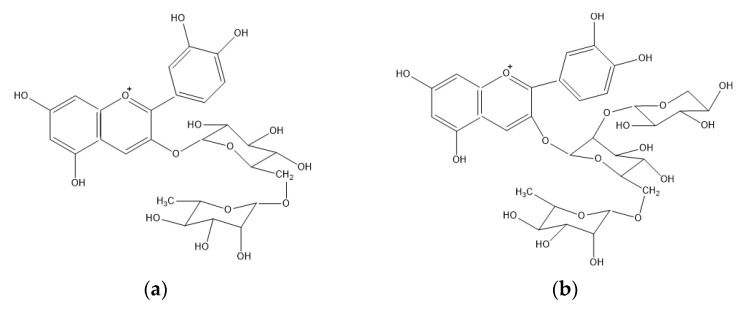
Chemical structure: (**a**) cyanidin 3-rutinoside; (**b**) cyanidin 3-xylosylrutinoside.

**Table 1 molecules-26-03904-t001:** Nutrient composition (value per 100 g fresh weight) [11].

Nutrient	Strawberry	Blackberry	Raspberry	Cranberry	Blueberry	Blackcurrant
Water (g)	90.95	88.15	85.75	87.32	84.21	83.95
Energy (kcal)	32	43	52	46	57	56
Protein (g)	0.67	1.39	1.2	0.46	0.74	1.4
Total lipid (fat) (g)	0.3	0.49	0.65	0.13	0.33	0.2
Carbohydrate (g)	7.68	9.61	11.94	11.97	14.49	13.8
Fiber, total dietary (g)	2	5.3	6.5	3.6	2.4	4.3
Sugars, total (g)	4.89	4.88	4.42	4.27	9.96	7.37
Calcium, Ca (mg)	16	29	25	8	6	33
Iron, Fe (mg)	0.41	0.62	0.69	0.23	0.28	1
Magnesium, Mg (mg)	13	20	22	6	6	13
Phosphorus, P (mg)	24	22	29	11	12	44
Potassium, K (mg)	153	162	151	80	77	275
Sodium, Na (mg)	1	1	1	2	1	1
Zinc, Zn (mg)	0.14	0.53	0.42	0.09	0.16	0.23
Copper, Cu (mg)	0.048	0.165	0.09	0.056	0.057	0.107
Selenium, Se (µg)	0.4	0.4	0.2	0.1	0.1	0.6
Vitamin C (mg)	58.8	21	26.2	14	9.7	41
Thiamin (mg)	0.024	0.02	0.032	0.012	0.037	0.04
Riboflavin (mg)	0.022	0.026	0.038	0.02	0.041	0.05
Niacin (mg)	0.386	0.646	0.598	0.101	0.418	0.1
Vitamin B6 (mg)	0.047	0.03	0.055	0.057	0.052	0.07
Folate, total (µg)	24	25	21	1	6	8
Vitamin A (µg)	1	11	2	3	3	2
Carotene, beta (µg)	7	128	12	38	32	25
Carotene, alpha (µg)	0	0	16	0	0	0
Lutein + zeaxanthin (µg)	26	118	136	91	80	47
Vitamin E (mg)	0.29	1.17	0.87	1.32	0.57	0.1
Vitamin K (phylloquinone) (µg)	2.2	19.8	7.8	5	19.3	11

**Table 2 molecules-26-03904-t002:** Contents of total phenolics and anthocyanins in berries (mg/100 g fresh weight).

Berries	Phenolics	Anthocyanins Contents
Strawberry (*Fragaria ananassa*)	317.2–443.4 [13]	32.6–52.4 [14]
	209.0–318.0 [15]	21.2–41.7 [16]
	264.0–324.0 [17]	32.0–36.0 [18]
Blackberry (*Rubus fructicosus*)	411.0–459.0 [17]	245.0–300.5 [16]
	417.8–555.2 [19]	114.4–241.5 [20]
	472.0–678.0 [21]	110.5–122.7 [19]
		143.0–211.0 [21]
Blueberry (*Vaccinium corymbosum*)	181.1–390.5 [22]	93.1–235.4 [22]
	261.9–585.3 [19]	94.5–301.0 [23]
	154.7–398.0 [23]	308.9–464.3 [16]
	212.7–460.4 [24]	143.5–822.7 [20]
	314.0–382.0 [17]	35.5–129.9 [19]
Cranberry (*Vaccinium macrocarpon*)	120.0–176.5 [25]	19.8–65.6 [25]
	163.4–315.9 [26]	111.5–168.5 [16]
		68.4–87.0 [27]
Raspberry (*Rubus idaeous*)	192.0–359.0 [28]	62.0–68.0 [29]
	505.0–529.0 [29]	19.0–51.0 [28]
	305.5–378.5 [30]	39.4–53.9 [14]
	305.8–503.9 [29]	72.4–111.8 [16]
	295.0–310.0 [17]	41.8–86.2 [31]
		68.0–80.0 [18]
Black Raspberry (*Rubus occidentalis*)	489.3–875.3 [30]	318.6–332.4 [31]
	970.0–990.0 [29]	585.0–593.0 [29]
	699.2–730.2 [31]	464.0–627.0 [21]
	890.0–1079.0 [21]	
Blackcurrant (*Ribes nigrum*)	498.0–1342.0 [21]	128.0–411.0 [21]
	817.0–1042.0 [32]	361.0–591.0 [16]
		233.4–237.8 [33]

**Table 3 molecules-26-03904-t003:** Contents of ellagic acid in berries (mg/100 g fresh weight).

Berries	Total Ellagic Acid After Hydrolysis	Free Ellagic Acid
Strawberry (*Fragaria ananassa*)	25.0–56.4 [13]	2.1–28.8 [13]
	19.3–48.3 [15]	0.6–2.6 [15]
	71.4–78.5 [14]	0.7–4.3 [14]
Blackberry (*Rubus fructicosus*)	30.0–33.8 [19]	ND ^1^
	10.6–51.5 [34]	
	35.7–54.7 [35]	
Blueberry (*Vaccinium corymbosum*)	0.8–6.7 [19]	ND ^1^
Cranberry (*Vaccinium macrocarpon*)	ND ^1^	ND ^1^
Raspberry (*Rubus idaeous*)	260.0–326.2 [14]	3.7–4.7 [14]
	83.9–210.4 [31]	2.0–5.5 [31]
	61.2–117.4 [36]	
	38.0–118.0 [28]	
Black Raspberry (*Rubus occidentalis*)	234.2–258.4 [31]	3.7–3.9 [31]
Blackcurrant (*Ribes nigrum*)	ND ^1^	ND ^1^

^1^ ND, not detected.

**Table 4 molecules-26-03904-t004:** Contents of PACs in berries, mg/100 g fresh weight.

Berries	Proanthocyanidins
Strawberry (*Fragaria ananassa*)	15.0–183.0 [42]
	34.2–57.0 [39]
	120.1–169.9 [43]
Blackberry (*Rubus fructicosus*)	5.0–46.0 [42]
	9.5–44.0 [43]
Blueberry (*Vaccinium corymbosum*)	311.0–335.0 [42]
	296.0–330.0 [39]
	318.0–346.0 [43]
Cranberry (*Vaccinium macrocarpon*)	646.5–691.3 [27]
	343.0–494.0 [43]
	399.0–412.0 [39]
Raspberry (*Rubus idaeous*)	76.9–80.6 [39]
Black Raspberry (*Rubus occidentalis*)	3.0–74.0 [42]
Blackcurrant (*Ribes nigrum*)	105.0–255.0 [42]
	120.6–165.8 [44]
	114.8–180.8 [43]

**Table 5 molecules-26-03904-t005:** Contents of flavonols in berries, mg/100 g fresh weight.

Berries	Flavonols
Strawberry (*Fragaria ananassa*)	1.8–5.6 [15]
	1.8–6.2 [13]
	0.8–1.6 [18]
	1.2–1.5 [46]
Blackberry (*Rubus fructicosus*)	10.2–16.0 [20]
	8.9–11.0 [19]
Blueberry (*Vaccinium corymbosum*)	15.0–17.0 [18]
	17.2–32.7 [20]
	19.4–23.8 [19]
	17.0–19.0 [47]
Cranberry (*Vaccinium macrocarpon*)	11.0–25.0 [48]
	15.7–26.3 [46]
	18.4–36.0 [49]
Raspberry (*Rubus idaeous*)	0.9–2.0 [49]
	0.6–0.8 [46]
	0.3–0.4 [47]
Black Raspberry (*Rubus occidentalis*)	10.3–19.0 [50,51]
Blackcurrant (*Ribes nigrum*)	12.5–15.0 [52]
	8.8–11.5 [46]

**Table 6 molecules-26-03904-t006:** ORAC value of berries.

Berry, Raw	ORAC Value, μmol TE/100 g
Strawberry	2154–8384 [104]
Red Raspberry	3748–5792 [104] 2220–2580 [29]
Blackberry	4686–7610 [104] 4160–7880 [21]
Blueberry	2746–9245 [104] 2627–6747 [24]
Blackcurrant	5010–10,144 [104] 4450–9200 [21]
Cranberry	8596–9679 [104]
Black raspberry	7470–7970 [29] 10,030–14,600 [21]

## Data Availability

Not applicable.

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
