# Peer review of "(untitled)"

_molecules, 2021, doi:10.3390/molecules26133904_

Round 1

Reviewer 1 Report

Berries are well documented popular foods with health functions. This manuscript presents a clear review of the recent progress of berry chemistry, bioactives and functions. The authors also summarized some proposed postulated mechanisms of berry phytochemicals. Overall, the manuscript is acceptable. 

Some concerns :
1. The challenges and perspectives of future research are not well stated in this review article.
2. Stilbenes are also frequently found in berries, I like the authors to present necessary information regarding these phytochemicals.

Author Response

Comments and Suggestions for Authors by reviewer 1:

Berries are well documented popular foods with health functions. This manuscript presents a clear review of the recent progress of berry chemistry, bioactives and functions. The authors also summarized some proposed postulated mechanisms of berry phytochemicals. Overall, the manuscript is acceptable.

1. The challenges and perspectives of future research are not well stated in this review article.

Ans: Future clinical trials are required to study and improve berries phenolic compound's bioavailability and extend the evidence that active compounds of berries can be used as medicinal foods against various diseases

2. Stilbenes are also frequently found in berries, I like the authors to present necessary information regarding these phytochemicals.

Ans: Thanks for the valuable comments, we definitely included the information stibenes.

Stilbenes are a specific class of non-flavonoid phenolic compounds present in berries. The most popular compound identified stilbenes in berries is resveratrol. Grapes and red wine are among the primary dietary sources of stilbenes. Stilbenes possess different biological and pharmacological activities potentially beneficial for human health, such as neuroprotective, antitumor, and antioxidant effects. Berries such as blueberry and cranberry contain stilbenes.

Reviewer 2 Report

The presented manuscript presents information gathered and published in a plethora of recent review articles, e.g., https://doi.org/10.1046/j.1365-2672.2001.01271.x, https://doi.org/10.1080/10408398.2011.608174, https://doi.org/10.3390/ijms161024673, https://doi.org/10.1007/s11130-010-0177-1, DOI:10.3233/MNM-140013, https://doi.org/10.1016/j.foodres.2012.11.004, https://doi.org/10.1016/j.jff.2017.07.050, https://doi.org/10.1039/C7FO01551H, DOI:10.5772/67104 and many more. Therefore a reason for having another review article dealing with the same subject may be highly questionable. The authors have had to provide why in fact they have entered in this project, but they failed to justify their reasons for this in the Introduction section. Furthermore, this section is rather poorly presented. No Introduction of a scientific publication should look this way, just listing the species tend to be presented and naming the main specialized compounds. The same is applicable for the Abstract.

Moreover, the main drawback of the presented review manuscript is that the authors have never provided information on how did they search the literature, which search engines were employed, what were the inclusion criteria, what were the keywords, which languages were searched and which were omitted, were PhD theses included or not and other criteria relevant to the study, which represent a common procedure in scientific reviews. 

Also, one major constraint is the language usage. The authors seem to have written the manuscript in rush, so many sentences are barely understandable. The manuscript should be meticulously edited by a native English speaker or a professional editing agency.

Botanical nomenclature is also questionable and the authors show lack of familiarity with it. The authors use italics for Latin names here-and-there with no particular norm. Species authority is discriminable used (it should not stand italicized), only for Ribes nigrum. Moreover, "sp." and "ssp." do not mean the same, neither should they be written irrespective of cursive style.

Author Response

Comments and Suggestions for Authors by reviewer 2:

1. The presented manuscript presents information gathered and published in a plethora of recent review articles, e.g., https://doi.org/10.1046/j.1365-2672.2001.01271.x, https://doi.org/10.1080/10408398.2011.608174, https://doi.org/10.3390/ijms161024673, https://doi.org/10.1007/s11130-010-0177-1, DOI:10.3233/MNM-140013, https://doi.org/10.1016/j.foodres.2012.11.004, https://doi.org/10.1016/j.jff.2017.07.050, https://doi.org/10.1039/C7FO01551H, DOI:10.5772/67104 and many more. Therefore a reason for having another review article dealing with the same subject may be highly questionable. The authors have had to provide why in fact they have entered in this project, but they failed to justify their reasons for this in the Introduction section. Furthermore, this section is rather poorly presented. No Introduction of a scientific publication should look this way, just listing the species tend to be presented and naming the main specialized compounds.

Ans: Thanks for the comments, most related review papers are about the prevention and biological activities. However, the main contents of this review paper is focused on the bioavailability and the clinical application, especially the therapeutic purpose. We revised the introduction and added some more clear information.

Functional plant-based foods (such as fruits, vegetables, berries) improve health, have a preventive effect, and diminish the risk of different chronic diseases in vivo and in vitro studies. Berries contain many phytochemicals, fibers, vitamins, and minerals. The primary phytochemicals in berry fruits are phenolic compounds, including flavonoids (anthocyanins, flavonols, flavones, flavanols, flavanones, and isoflavonoids), tannins, and phenolic acids. [1]. Berries have high levels of polyphenols, thus it is possible to use them for treating various diseases pharmacologically by acting on oxidative stress and inflammation, which are often the leading causes of diabetes, neuro-logical, cardiovascular diseases, and cancer. This review examines commonly consumed berries: blackberry (Rubus sp.), blackcurrant (Ribes nigrum), blueberry (Vaccinium sp.), cranberry (Vaccinium macrocarpon), raspberry (Rubus idaeus), black raspberry (Rubus occidentalis), and strawberry (Fragaria ananassa) and their polyphenols as potential medicinal foods (due to the presence of pharmacologically active compounds) in the treatment of diabetes, cardiovascular and other diseases. Biologically active components of berries possess antioxidant, antihyperlipidemic, antihypertensive, antiproliferative effects, anti-inflammatory, antibacterial, and antiviral responses [2]. Moreover, much attention is on the bioavailability of the active berry components. Increasing bioavailability decreases the number of biotransformations of active compounds into the gastrointestinal tract and improves the health benefits of berries. Hence, this comprehensive review shows that berries and their bioactive compounds possess medicinal properties and therapeutic potential.

The same is applicable for the Abstract.

Revised abstract.

Functional plant-based foods (such as fruits, vegetables, berries) improve health, prevent diseases, and diminish the risk of different chronic diseases in vivo and in vitro studies. Berries contain many phytochemicals, fibers, vitamins, and minerals. The primary phytochemicals in berry fruits are phenolic compounds, including flavonoids (anthocyanins, flavonols, flavones, flavanols, flavanones, and isoflavonoids), tannins, and phenolic acids. Berries have high levels of polyphenols, thus it is used to treat various diseases pharmacologically by combating oxidative stress and inflammation, which are often the leading causes of diabetes, neurological, cardiovascular diseases, and cancer. This review examines commonly consumed berries: blackberry (Rubus sp.), blackcurrant (Ribes nigrum), blueberry (Vaccinium sp.), cranberry (Vaccinium macrocarpon), raspberry (Rubus idaeus), black raspberry (Rubus occidentalis), and strawberry (Fragaria ananassa) and their polyphenols as potential medicinal foods (due to the presence of pharmacologically active compounds) in the treatment of diabetes, cardiovascular and other diseases. Moreover, much attention is on the bioavailability of active berry components. Hence, this comprehensive review shows that berries and their bioactive compounds possess medicinal properties and therapeutic potential. Nevertheless, future clinical trials are needed to confirm the bioavailability of berries phenolics and extend the evidence, for which berries can be used as medicinal foods against various diseases.

2. Moreover, the main drawback of the presented review manuscript is that the authors have never provided information on how did they search the literature, which search engines were employed, what were the inclusion criteria, what were the keywords, which languages were searched and which were omitted, were PhD thesis included or not and other criteria relevant to the study, which represent a common procedure in scientific reviews.

Ans: The authors of this comprehensive review article carried out a literature search for relevant articles regarding functional and pharmacological activities of berries by determining sourced or literature in the form of primary data or official books, national or international journals published till May 2021. Additionally, data searches were also conducted using different online platforms. During writing this review article, the main references were cited from the trusted source, such as Medline (PubMed), Scopus, Google Scholar, NCBI, Science Direct, ResearchGate, Web of Science and other trusted journals publishes with keywords: "berries", "phytochemicals", "bioavailability", "pharmaceuticals properties", "health benefits". This review article does not have any inclusion criteria, also PhD theses not included in the review, the search was only limited to articles published in the English language.

3. Also, one major constraint is the language usage. The authors seem to have written the manuscript in rush, so many sentences are barely understandable. The manuscript should be meticulously edited by a native English speaker or a professional editing agency.

Ans: The whole manuscript was cross-checked for grammatical errors by native English speakers, and all the changes were highlighted in yellow.

4. Botanical nomenclature is also questionable and the authors show lack of familiarity with it. The authors use italics for Latin names here-and-there with no particular norm. Species authority is discriminable used (it should not stand italicized), only for Ribes nigrum. Moreover, "sp." and "ssp." do not mean the same, neither should they be written irrespective of cursive style.

Ans: The whole manuscript has been revised, and botanical nomenclature corrected.

Reviewer 3 Report

This impressive review is heroic in the mode of reviews of the past that served as collections of all pertinent references. I can’t speak to the completeness of the reference list, but I can certainly say that the review appears to be extremely well researched by authors who are experts in the field. In general, the submission is well written but could use substantial English language editing. Some examples are: 1) singular / plural noun and verb agreements are often askew, 2) verbs are missing from many sentences.

Page 2 Line 42 – Need to indicate somewhere in Table 1 that values are per 100g of berry.

Page 4 Line 57 – Why only show the structure of cyanidin? Why not the other compounds as well?

Page 4 Line 61 – The compound shown in Figure 1 is called cyanidin in the narrative but it’s called cyanidin anthocyanidin in the figure legend. Which one is correct? Is it because it’s from the anthocyanin family? The distinction between anthocyanin and anthocyanidin should be made clear. Perhaps note that it is responsible for the reddish purple color of these berries?

Page 4 Line 68 – Figure 2, structure (a) appears pixelated.

Page 7 Line 142 – The following sentence is not clear to me, especially the word ‘splited’. “About 73% of anthocyanins consumed from black currant enter the colon and splited by microorganisms.”

Page 8 Lines 211-215 – I believe that the authors are saying that methods to improve bioavailability of phenolic compounds are needed, but this idea is not clearly expressed in this sentence.

Page 9 Lines 234-238 – It would be good to see the structures of these anti-cancer compounds.

Page 9 Lines 252-253 – It isn’t clear what the authors are trying to say in this sentence.

Page 10 Lines 268-271 - It isn’t clear what the authors are trying to say in this sentence.

Page 11 Lines 342-343 – It doesn’t seem possible to gavage 250 g of anything to a mouse. Maybe the unit isn’t correct?

Page 13 Lines 425-426 – “by suppressing free radicals by donating hydrogen molecules”… I think that the authors mean to say hydrogen ‘atoms’ (H), not hydrogen ‘molecules’ (H2).

Page 13 Lines 445-446 - It isn’t clear what the authors are trying to say in this sentence. Who or what are the ‘elders’?

Author Response

Comments and Suggestions for Authors by reviewer 3:

1. This impressive review is heroic in the mode of reviews of the past that served as collections of all pertinent references. I can’t speak to the completeness of the reference list, but I can certainly say that the review appears to be extremely well researched by authors who are experts in the field. In general, the submission is well written but could use substantial English language editing. Some examples are: 1) singular / plural noun and verb agreements are often askew, 2) verbs are missing from many sentences.

Ans: The whole manuscript was cross-checked for grammatical errors by native English speakers, and all the changes were highlighted in yellow.

2. Page 2 Line 42 – Need to indicate somewhere in Table 1 that values are per 100g of berry.

Ans: It is indicated in Table 1 as the comments.

3. Page 4 Line 57 – Why only show the structure of cyanidin? Why not the other compounds as well?

Ans: The structures of anthocyanidins is revised by the valuable comment.

Figure 1. Chemical structure of anthocyanidins: R1=H, R2=H Pelargonidin; R1=OH, R2=H Cyanidin; R1=OH, R2=OH Delphinidin; R1=OCH3, R2=H Peonidin; R1=OCH3, R2=OH Petunidin; R1=OCH3, R2=OCH3 Malvidin.

4. Page 4 Line 61 – The compound shown in Figure 1 is called cyanidin in the narrative but it’s called cyanidin anthocyanidin in the figure legend. Which one is correct? Is it because it’s from the anthocyanin family? The distinction between anthocyanin and anthocyanidin should be made clear. Perhaps note that it is responsible for the reddish purple color of these berries?

Ans: Corrected name Figure 1 – Chemical structure of anthocyanidins, we modified it accordingly.

Cyanidin is an anthocyanidin and the main plant pigment found in many red berries, including blueberries, blackberries, blueberries, cranberries, and raspberries. Anthocyanins are glycoside by linking sugars with anthocyanidins.

5. Page 4 Line 68 – Figure 2, structure (a) appears pixelated.

Ans: We modified it accordingly.

Figure 2. Chemical structure A-type proanthocyanidins

6. Page 7 Line 142 – The following sentence is not clear to me, especially the word ‘splited’. “About 73% of anthocyanins consumed from black currant enter the colon and splited by microorganisms.”

Ans: This sentence is revised: After consumed black currant, about 73% of its anthocyanins enter the colon and metabolize by microorganisms.

7. Page 8 Lines 211-215 – I believe that the authors are saying that methods to improve bioavailability of phenolic compounds are needed, but this idea is not clearly expressed in this sentence.

Ans: This sentence is revised as: The bioavailability of phenolic compounds is needed to improve by using various methods of increasing the stability and solubility in the gastrointestinal tract, such as selective inclusion, solid dispersion, phospholipid liposomes, microemulsion technology, and the conversion of flavonoid aglycones into nanoparticles.

8. Page 9 Lines 234-238 – It would be good to see the structures of these anti-cancer compounds.

Ans: The structures these anti-cancer compounds are added (Figure 5)

(a)

(b)

Figure 5. Chemical structure: (a) cyanidin 3-rutinoside; (b) cyanidin 3-xylosylrutinoside.

10. Page 9 Lines 252-253 – It isn’t clear what the authors are trying to say in this sentence.

Ans: The sentence is revised as: The antioxidant properties of black currant are mainly associated with anthocyanins.

11. Page 10 Lines 268-271 - It isn’t clear what the authors are trying to say in this sentence.

Ans: The sentence is revised as: Consumption of cranberry juice has improved plasma antioxidant capacity, and thereby decreased the circulating concentrations oxidized LDL-c in women with metabolic syndrome as well as decreased blood markers of oxidative stress in healthy volunteers and patients with cardiovascular risk factors.

12. Page 11 Lines 342-343 – It doesn’t seem possible to gavage 250 g of anything to a mouse. Maybe the unit isn’t correct?

Ans: It’s a type error, and is revised as 250 µg.

13. Page 13 Lines 425-426 – “by suppressing free radicals by donating hydrogen molecules”… I think that the authors mean to say hydrogen ‘atoms’ (H), not hydrogen ‘molecules’ (H2).

Ans: It’s a type error and is revised as “atoms”

14. Page 13 Lines 445-446 - It isn’t clear what the authors are trying to say in this sentence. Who or what are the ‘elders’?

Ans: This is revised as “elderly”.

Round 2

Reviewer 2 Report

The authors have rewritten Abstract and the Introduction section. It is more clear this way. However, they reprised the same sentences and phrases in both chapetrs, which is rather not particularly welcome in scientific literature.

Searching for literature is vaguely elaborated. Keywords used are stated ad hoc (i.e., if one searches for articles that contain "phytochemicals", a bunch of them will return that were not cited in this review). I strongly encourage the authors to consult PRISMA checklist (http://www.prisma-statement.org/PRISMAStatement/Checklist), which is usually employed for manuscripts containing meta-analysis. 

Language usage is not improved. The authors might wanted to send the manuscript to a native English speaker or a professional editing agency for proofreading.

Author Response

The authors have rewritten Abstract and the Introduction section. It is more clear this way. However, they reprised the same sentences and phrases in both chapetrs, which is rather not particularly welcome in scientific literature.

Searching for literature is vaguely elaborated. Keywords used are stated ad hoc (i.e., if one searches for articles that contain "phytochemicals", a bunch of them will return that were not cited in this review). I strongly encourage the authors to consult PRISMA checklist (http://www.prisma-statement.org/PRISMAStatement/Checklist), which is usually employed for manuscripts containing meta-analysis.

Ans: Thank you so much for valuable comments. We changed the introduction and added some information, also include several meta-analysis.

Many studies and reviews have reported the relationship between fruit intake and health. Some berries use as ingredients in functional foods and dietary supplements.  Berries are rich in nutrients, and phytochemicals can improve health and prevent various chronic diseases in vivo and in vitro studies. The primary phytochemicals in berry fruits are phenolic compounds, including flavonoids (anthocyanins, flavonols, flavones, flavanols, flavanones, and isoflavonoids), tannins, and phenolic acids [1].

Raimundo and his co-workers [2] conducted a meta-analysis of a considerable number of different human randomized clinical trials to estimate the effects of polyphenol intake on biomarkers (such as the level of glucose, insulin, and others) in people with prediabetes and T2D. They found that the consumption of polyphenols may contribute to lower glucose levels. 

Other meta-analysis of 128 randomized clinical trials carried to investigate the effects of plant sources of anthocyanins and ellagitannins (berries, nuts, red grapes/wine) on cardiometabolic risk biomarkers. Both anthocyanin- and ellagitannin-containing products reduced total cholesterol. However, blood pressure was significantly decreased by the sources of anthocyanins such as berries and red grapes/wine. In contrast, the ellagitannin-products, especially nuts, most reduced waist circumference, LDL-cholesterol, triglycerides, and glucose considerably [3].

Berries have a high concentration of polyphenols, thus it is possible to use them for treating various diseases pharmacologically by acting on oxidative stress and inflammation, which are often the leading causes of diabetes, neurological, cardiovascular diseases, and cancer.

Blueberries have antioxidant and anti-inflammatory effects and also possess neurocognitive benefits. Consumption of blueberry juice improved memory function in older adults with early memory decline [4]. Black raspberries are sources of phenolic compounds like ellagic acid and anthocyanins that have potential cancer chemopreventive activity, confirmed human clinical trials [5,6]. Blackcurrant powder reduced the activity of some colon cancer markers by acting as prebiotic agents [7]. Antioxidant and anti-inflammatory properties of strawberry displayed due to the high content of bioactive compounds, such as vitamins and phenols, in several in vitro and in vivo studies [8,9].

This review examines commonly consumed berries: blackberry (Rubus sp.), black-currant (Ribes nigrum), blueberry (Vaccinium sp.), cranberry (Vaccinium macrocarpon), raspberry (Rubus idaeus), black raspberry (Rubus occidentalis), and strawberry (Fragaria ananassa) and their polyphenols as potential medicinal foods (due to the presence of pharmacologically active compounds) in the treatment of various diseases and disorders. Biologically active components of berries possess antioxidant, antihyperlipidemic, antihypertensive, anti-proliferative effects, anti-inflammatory, antibacterial, and antiviral responses [10].

Polyphenols have a low bioavailability, and increasing bioavailability reduces the number of biotransformations of active compounds into the gastrointestinal tract and improves the health benefits of berries. This review discusses the studies conducted in vivo, which consider the berry's polyphenol bioavailability. Hence, this comprehensive review shows that berries and their bioactive compounds possess medicinal properties and therapeutic potential.

Language usage is not improved. The authors might wanted to send the manuscript to a native English speaker or a professional editing agency for proofreading.

Ans: Thank you for comments, this manuscript has sent to a native English speaker for proofreading.